# Specificities and Dynamics of Transposable Elements in Land Plants

**DOI:** 10.3390/biology11040488

**Published:** 2022-03-23

**Authors:** Corinne Mhiri, Filipe Borges, Marie-Angèle Grandbastien

**Affiliations:** Université Paris-Saclay, INRAE, AgroParisTech, Institut Jean-Pierre Bourgin (IJPB), 78000 Versailles, France; filipe.borges@inrae.fr (F.B.); marie-angele.grandbastien@inrae.fr (M.-A.G.)

**Keywords:** transposable element, transposition control, plant genome, TE classification, evolution, stress, silencing, epigenetic, methylation, polyploidy

## Abstract

**Simple Summary:**

Transposable elements are dynamic components of plant genomes, and display a high diversity of lineages and distribution as the result of evolutionary driving forces and overlapping mechanisms of genetic and epigenetic regulation. They are now regarded as main contributors for genome evolution and function, and important regulators of endogenous gene expression. In this review, we survey recent progress and current challenges in the identification and classification of transposon lineages in complex plant genomes, highlighting the molecular specificities that may explain the expansion and diversification of mobile genetic elements in land plants.

**Abstract:**

Transposable elements (TEs) are important components of most plant genomes. These mobile repetitive sequences are highly diverse in terms of abundance, structure, transposition mechanisms, activity and insertion specificities across plant species. This review will survey the different mechanisms that may explain the variability of TE patterns in land plants, highlighting the tight connection between TE dynamics and host genome specificities, and their co-evolution to face and adapt to a changing environment. We present the current TE classification in land plants, and describe the different levels of genetic and epigenetic controls originating from the plant, the TE itself, or external environmental factors. Such overlapping mechanisms of TE regulation might be responsible for the high diversity and dynamics of plant TEs observed in nature.

## 1. Introduction

Transposable elements (TEs) can be defined as repetitive DNA sequences able to move/transpose throughout their host genome. They were first discovered in maize by Barbara McClintock in the 1940s as controlling elements able to modify gene expression and change their location upon genomic stress, such as chromosomal double-strand breaks [1]. With the development of molecular biology and sequencing technologies, the detection of mobile genetic elements has been generalized to almost all living organisms. Their high abundance in some genomes (>80% in maize) [2] and extreme diversity in transposition modes and insertion profiles resulted in a progressive interest of the research community for studying the biology of these repetitive sequences, and the way they interact and coevolve with their host genome. Initially seen as invading parasitic sequences because of their proliferative and mutational abilities [3,4], TEs have been progressively considered as important components of eukaryotic genomes since the discovery of some ‘useful’ TEs contributing to gene expression regulation or enzymatic functions [5,6]. Such opinion changes, considering TEs as ‘facilitators of evolution’ for their host organisms, have been well documented [7,8,9,10].

After presenting a general updated picture of TE composition and diversity in plants, this review will also focus on the different control levels acting in plants to regulate TE expression, activity and expansion, thus contributing to plant evolution and adaptation.

## 2. Plant TE Landscape

### 2.1. A Highly Variable TE Abundance

Due to their ubiquity, abundance and transposition activity, TEs have been proposed as major contributors to genome size, along with other mechanisms such as recombinational rate and polyploidization [11]. Among living organisms, land plants (regrouping Bryophytes, Pteridophytes, and seed plants), and especially flowering plants (Angiosperms), display one of the largest genome size variability exceeding 2400-fold, with C-values ranging from 0.07 pg (65 Mb/1 C) for the small carnivorous plant *Genlisea tuberosa* to 152.23 pg (149 Gb/1 C) for the monocot lily species *Paris japonica* (https://cvalues.science.kew.org/ accessed on 10 March 2022). Indeed, TEs seem to account for the variable proportion of plant genomes sequenced to date, spanning from ~3% in the small 82 Mb carnivorous *Utricularia gibba* [12] to ~85% in allohexaploid wheat (*Triticum aestivum*) [13] or maize genome [2].

### 2.2. Challenging Evaluation of Plant TE Diversity and Classification

Our knowledge about TEs structure, organization and transposition mechanisms has greatly progressed since the discovery of the first TE sequences in plants (the maize Ac/Ds elements) in 1984 by two different laboratories [14,15]. Transposable elements are extremely diverse and use various mechanisms to move. The so-called autonomous elements encode all specific functions to achieve their mobility, while some non-autonomous elements hitch-hike mobility proteins from autonomous copies, or from other TEs to transpose.

Several types of TEs might co-exist in a given genome, and each TE type can harbor multiple TE copies clustered into different families according to their sequence similarity. As some transposition mechanisms are prone to generate mutations, each family might have evolved over time, displaying a continuum of more or less diverged copies composed of both autonomous and defective elements [7,8,10,16]. Such behaviour made TE identification and classification a difficult task.

In 1989, the first TE nomenclature organized TEs into two classes according to their transposition intermediate [17]: (1) an RNA intermediate for retrotransposons (class I elements) that move via a replicative “copy-and-paste” mechanism, where a “mother” copy gives rise to several “daughter” copies without excising itself; (2) a DNA intermediate for DNA transposons (class II elements) that use a conservative “cut-and-paste” mechanism for their transposition, where the “mother” copy excises from its location to insert elsewhere in the genome (“jumping gene” concept).

This bimodal schematic classification has been further refined in 2007 (1) by creating subclasses in the class II group to include TEs that use a DNA replicative mechanism, such as Helitrons, and (2) by setting the “80-80-80” rule to define the identity percentage TE copies should share to belong to the same family, i.e., sequences over 80 bp sharing at least 80% sequence identity in at least 80% of their internal domain or terminal repeats (or both) [16,18,19]. This TE hierarchical classification splits TEs into the two previously cited classes (Class I and Class II), then into subclasses, orders and superfamilies [18]. This classification is based on the presence and order of coding regions for specific proteins or structural motifs present in TE sequences, and on their transposition specificities (presence and sequence of target site duplications (TSDs), i.e., short 2–11 bp DNA duplications, generated upon TE transposition at the new insertion site). For example, the presence of long terminal repeats (LTRs) in direct orientation at TE 3′ and 5′ ends is the signature of LTR-retrotransposons (LTR-RTs). The occurrence of the reverse transcriptase (RT) domain in TE ORF is a hallmark of most but not all Class I retrotransposons. Like self-replicating entities as viruses, TEs have a modular evolution, exchanging essential or facultative protein-coding domains that may blur TE classification [19].

If these first four classification levels are generally well accepted, the need of extra levels to reach the TE family level is still unclear and may vary according to the TE considered. For example, the superfamily subgroups “chromovirus” or “non-chromovirus” have been introduced for Gypsy LTR-RTs [20]. An increase in family/lineage number or even new superfamilies will probably arise with the accumulation of genome sequencing data and the improvement of bioinformatic pipelines for TE detection and annotation. This hierarchical classification also does not include some non-autonomous TEs, i.e., TEs that are still able to transpose but need extra-function in trans coming from another TE element (phylogenetically related or not). This includes LARDs (large retrotransposon derivatives), TRIMs (transposon in miniature) and SMARTs (small LTR-retrotransposon) for Class I TEs, and MITEs (miniature inverted-repeats transposable elements), SNACs (small non-autonomous CACTA) or MULEs (Mu-like elements) for class II TEs. Many such non-autonomous elements share enough sequence similarity/motifs to be easily linked to a candidate “helper” autonomous TE family, as reported in rice with the isolation of the complete RIRE2 Gyspy retrotransposon displaying a high LTR sequence similarity to sequence extremities of the defective *Dasheng* retrotransposon [21]. Another example is the description of both complete and truncated—but nevertheless active—CACTA *Caspar* elements in Triticeae [16,22]. Some other non-autonomous TEs may not present any clear feature allowing them to be classified, as some non-autonomous short TIR-harboring Tes that may share only a few bases homology with the autonomous helper [16]. In the classification we propose, we choose to include the non-autonomous SINES as a full order, as these TEs do not correspond to deleted versions of autonomous class I TEs.

Table 1 and Figure 1 present, respectively, the up-to-date classification and structures of plant transposable elements adapted from Wicker et al. [16]. It is important to note that only a subset of existing TE superfamilies in all living organisms (as reported in repbase https://www.girinst.org/repbase/update/browse.php accessed on 10 March 2022) has been detected in land plants (~30% of class II superfamilies, and ~17% of class I—representing 20% of all described superfamilies). Plant TEs fall into six different orders, four orders corresponding to class I elements (LTR-RTs, *Penelope*-like elements (PLE), long interspaced nuclear elements (LINEs), short interspaced nuclear elements (SINEs)) and two orders including elements of class II (terminal inverted repeat (TIR) transposons, Helitrons) (Table 1).

In plants, class II elements (subclass 1) belonging to the TIR order (also called DNA transposons) fall into six superfamilies, based on the structure and sequence of their transposase and on the sequence of their terminal inverted repeats (TIRs) (Figure 1). Transposase is the protein catalyzing their transposition, while TIRs harbor key sequences recognized by the DNA-binding domains of the transposase during a transposition event. Some TIR elements also harbor additional coding sequences, as the maize MuDR, and plant CACTA or PIF/Harbinger elements [16]. Most of these superfamilies are also characterized by specific target site duplications (TSDs) lengths, generated after the filling of DNA nicks generated by the transposase on the integration site. Their transposition is not always strictly conservative and could lead to an increase of copy number if it occurs before a DNA replication fork [32].

Replicating plant TEs fall into two major groups: (1) Helitrons (class II-subclass 2, see Table 1) replicate through a rolling-circle (RC) mechanism from one DNA strand, without generating TSDs, by using a RepHel protein with a RC replication initiator (Rep) and DNA helicase (Hel) domains, in association with an ssDNA-binding “replication protein A” (RPA) [33]. (2) Class I retrotransposons replicate from RNA templates by reverse transcription using a TE-encoded reverse transcriptase (RT) and use at least one additional protein to mediate their insertion into their host genome, such as endonuclease (EN) or DDE integrase (INT). We do not include the DIRS superfamily as a member of land plants, as DIRS elements have only been found in green algae until now [34].

Among the four retrotransposon orders present in plants, SINEs occupy a particular place, as these small non-coding and non-autonomous elements of a few hundred base pairs exploit the transposition machinery of LINEs to ensure their amplification. Plant SINEs are derived from tRNAs [28]. They are transcribed by polymerase III, harbor short degenerated internal promoters (A and B boxes), and display mostly A tail at the 3′-end. Apart from these small structural domains, SINEs display a high sequence diversity that hinders their detection and characterization. Recently, a 37 pb Angio-domain located in the 3′-end has been reported in many Angiosperm SINEs [29].

The second non-LTR retrotransposon order present in plants, LINEs, contains elements belonging to the L1 and the RTE (retrotransposable element) superfamilies (Table 1 and Figure 1), which are two of the five known superfamilies of LINEs detected in eukaryotes [16]. RTE and L1 LINEs have one or two open reading frames (ORFs), respectively, and code for proteins required for retrotransposition, such as an endonuclease (EN), a RT, and often a ribonuclease H (RNase H (RH)). The L1 ORF1 is involved in the binding, protection and transport of the RNA intermediate used for retrotransposition. At their 3′-end lies a stretch of (A)_n_ for L1 or (GTT)_n_ for RTE involved in the reverse transcription initiation. A recent study shows that plant LINEs extracted from 23 genomes fall into only seven L1 and one RTE families/lineages/subclades [27]. As the reverse transcription starts from the 3′-end of LINEs and does not always reach the 5′-end, many incomplete daughter copies can be generated.

Between their bordering direct repeats 5′-LTR and 3′-LTR, autonomous LTR-retrotransposons (LTR-RTs) code for structural capsid-like (GAG) and functional (POL) proteins needed for their retrotransposition cycle (RT = reverse transcriptase, RH, INT = integrase), resembling the replication cycle of retroviruses. Only two out of the five superfamilies found in eukaryotes are represented in plants [16]. Plant LTR-RTs are further classified into Copia/Ty1 or Gypsy/Ty3 superfamilies according to the order of their coding *pol* domains. Recently, a systematic survey of plant LTR-RTs in 80 plant genomes refined the classification by introducing 16 lineages/families into the Copia/Ty1 superfamily and 14 lineages/families into the Gypsy/Ty3 group (six with a chromo-domain and eight without) [20]. Two Gypsy lineages, Chlamyvir and Tcn1, having only representatives in algae and non-*Viridiplantae* species, have not been included in Table 1. Non-autonomous derivatives of variable size (from a few hundred bp up to 25 kb) have been characterized in plants, containing between both LTRs a DNA sequence of variable length, either non-coding or reminiscent of some retrotransposon internal domains. Large internal sequences (>4 kb) define LARDs (large retrotransposon derivatives), and short ones (<4 kb) are often called TRIM (terminal repeat retrotransposon in miniature) [16].

Retrotransposons belonging to the *Penelope*-like elements (PLE) order are also found in some plant genomes, but with a patchy distribution. PLE encode an RT domain related to telomerase, a highly specialized class of non-mobile RTs responsible for chromosome end maintenance in most eukaryotes. Some PLEs also carry a second EN domain with a specific GIY-YIG motif. PLEs are bordered by repeats in direct or reverse orientation and are often subjected to 5′ truncation upon retrotransposition, as non-LTR retrotransposons. EN(+)PLEs (*Dryads* elements belonging to the *Penelope*/*Poseidon* group) and EN(-)PLEs have been found in some Conifer genomes (Table 1, Figure 1), and some of them were presumably derived from a horizontal transfer (HT) event [26].

The accumulation of sequenced genomes and TE detection pipelines allow the analysis and comparison of TE composition and diversity across plant genera. Figure 2 presents a heatmap of genomic percentage of four types of TEs—LTR-RTs, LINEs, SINEs and TIR DNA transposons—across 74 Angiosperm species displaying variable genome sizes (Data collected from [35] in the Supplementary Tables S1 and S2 of this article). Among the different types of TEs, LTR-RTs (Copia/Ty1 and Gypsy/Ty3) occupy the largest proportion of these genomes, the highest being up to 80% in *Zea mays*. Such an increase can result from rapid amplification of only a few families. For example, *Oryza australiensis* has undergone a recent burst of transposition involving only three families (one Copia = RIRE1 and two Gypsy = Wallabi and Kangourou), which compose 60% of its genome [36]. Genomes of the legume tribe *Fabeae* are also dominated by the Ty3/Gypsy Ogre family/lineage, that accounts for 57% of genome size variation on average in this clade [37]. The predominance of one type of LTR-RT may vary depending on the taxa considered. For example, in *Gossypium* species, Copia LTR-RTs have accumulated in the small genome of *G. raimondii* (880 Mb), while Gypsy LTR-RTs (mainly Gorge3) have proliferated in large genomes lineages of *G. herbaceum* 1667 Mb) and *G. exiguum* (2460 Mb) [38]. Plant species belonging to the same order (see Brassicales, Poales, Figure 2) might display different TE compositions and genome sizes. Some plant species also harbor specific TE composition as shown in Figure 2, with dominance of LINEs retrotransposons or TIR DNA transposon for the aquatic plant coontail *Ceratophyllum demersum* (33.6% of total genome size) and the small herbaceous plant *Trichopus zeylanicus* (~27.3% of total genome size). Non-LTR-retrotransposons have been shown to be abundant (~11.7%) in other plant genomes, such as *Arachis ipaenis*, one of the peanut parental genomes [39].

What are the forces taking place in the generation of such variable TE patterns in plants? The following section will present some of the factors known to influence TE diversity and dynamics in plants at the molecular level.

## 3. Regulating Factors of TE Transposition Control at the Molecular Level

The origin of species-specific differences in TE composition is still poorly understood. The TE profile observed in a given plant species may be the result of multiple regulating sources coming from TE themselves (regulatory motifs, biology) and host plant characteristics (epigenetic control, genome size, polyploidy level, sequence elimination mechanisms and reproductive system). All these evolutionary forces combined may lead to an equilibrium status, as illustrated in Figure 3 and discussed in the following paragraphs.

### 3.1. TE Regulatory Motifs Involved in TE Transcription/Activation

Despite the need for host cell machinery, autonomous TEs express specific proteins to ensure their self-propagation. These elements harbor regulatory motifs to promote their transcription and regulation, such as promoters for RNA polymerase II (Pol II, or Pol III for SINEs), and polyadenylation signals. The location of such cis-regulating sequences depends on the TE considered, and may disappear in truncated non-autonomous TEs [40]: in LTR-RTs, such motifs are present in duplicates in both 5′- and 3′-LTRs (and are also conserved in solo-LTRs, i.e., LTR-RT deleted versions generated upon ectopic recombination between LTRs); in class II TIR DNA transposons, such sequences are located in the 5′ part of the coding sequence between TIRs; in LINEs, they are located in the 5′-UTR (untranslated region) and in the 3′-UTR, and only in the 5′-UTR in SINEs (Pol III promoter). Transcription regulation is a key point, especially for class I TEs that use transcripts not only for protein synthesis, but also as a matrix for their reverse transcription. The analysis of transcriptional activity of 56 families of transposable elements in different maize organs and cell culture reveals a higher expression of Gypsy LTR-RTs compared to other classes. In maize, expression profiles in organs vary according to the type of transposable elements, and seems to originate from a few active copies [41].

Since the first report in 1985 of transposition of the maize Bs1 element following barley stripe mosaic virus infection [42], the activation of transposable elements under various environmental challenges has been extensively reviewed [43]. In sessile organisms like plants, the transcriptional activation of TEs has been reported to be induced by a wide array of activating stimuli including biotic and abiotic stresses, such as pathogens and their elicitors, tissue culture, wounding, temperature or drought, UV or X-ray irradiation [43,44,45]. Different regulatory motifs behaving as transcriptional activators and involved in TE transcriptional activation have been isolated in the LTR of active tobacco Tnt1 or Tto1 LTR-RTs [46,47,48,49]. Such cis-regulating elements present in LTRs may vary within a single retrotransposon family and may favor the transcription (first step of transposition) of some specific subfamilies depending on the type of stress applied, as exemplified for tobacco Tnt1 LTR-RTs subfamilies [50]. Such fine-tuned transcriptional regulation upon stress may favor the transposition and thus the impact of some TEs on plant evolution over time. Comparative studies of Tnt1-related elements in wild tomato species revealed a similar fast evolution of LTR regulatory sequences, as well as for the *Citrus sinensis* Tc1 and Tc2 LTR-RTs [43].

### 3.2. Epigenetic Control of TEs

In order to prevent the potentially deleterious consequences of excessive transposition, host organisms have developed several mechanisms to repress (silence) expression of their endogenous TEs. Such processes of heritable inactivation of gene expression without altering the DNA sequence belong to the domain of epigenetics.

Phenotypes associated with *Ac* or *Spm* transposon silencing have been extensively described in maize in the early 1960s by Barbara McClintock and other researchers (reviewed in [51]). TEs can be found in multiple epigenetic states, ranging from a transcriptionally active state leading to transposition, to a silenced state without any transcriptional activity. Such silencing can be family-specific [52] and heritable throughout generations, and may also depend on the genomic context of the insertion site (near/within gene, pericentromeric or centromeric location) [53].

For example, de novo silencing of a newly inserted autonomous TE (as represented in Figure 3, step 1) initiates generally at the post-transcriptional level (post-transcriptional gene silencing (PTGS)) and results in degradation of TE mRNAs or interference with their translation (Figure 3-step2). Then, a second mechanism ensures that TE silencing is maintained at the transcriptional level (transcriptional gene silencing (TGS)), by blocking TE transcription via DNA methylation and chromatin modifications at the corresponding TE sequence (Figure 3-step3). Such pathways involve different players including 21- to 24-nucleotide (-nt) small interfering RNAs (siRNAs), cytosine methylation (in CG, CHG or CHH contexts, where H = A, T or C), histone methylation (mainly di-methylation on lysine 9 of histone H3-H3K9me2) and recruitment of numerous host-encoded proteins. These epigenetic silencing events have been thoroughly studied in the model plant *Arabidopsis thaliana* (for a recent review, see Zhang et al. [54]).

One of the key events to initiate epigenetic silencing is the production of double-stranded RNAs from abundant, coding or non-coding aberrant RNAs, that occurs following hairpin formation, partial hybridization of two complementary mRNA, or produced by RNA-DEPENDENT RNA POLYMERASES (as RDR2/RDR6 in *Arabidopsis*) [55]. Depending on their type and origin, double-stranded RNAs are cleaved by DICER-LIKE PROTEINS into siRNAs of different sizes (21/22-nt by DCL2/4, 24-nt by DCL3 in *Arabidopsis*), and loaded onto ARGONAUTE proteins (AGO1 to AGO10 in *Arabidopsis*) to target complementary RNAs [55]. Several silencing outcomes may occur depending on the silencing complex interacting with the paired AGO/siRNA considered: for example, the RNA-induced silencing complex (RISC) associated with the paired AGO1/21–22-nt siRNA or microRNAs (miRNAs) will target and cleave complementary mRNAs, triggering PTGS. In contrast, TGS requires the plant-specific Pol IV to produce short non-coding RNAs that are converted into double-stranded RNA by RDR2, and processed into 24-nt siRNA by DCL3. These siRNAs are loaded into AGO4/6 to target complementary non-coding transcripts generated by another plant-specific RNA polymerase (Pol V), and recruit the DNA methyltransferase DOMAINS REARRANGED METHYLASE 2 (DRM2) that catalyzes de novo DNA methylation, leading to TGS. This mechanism is known as RNA-directed DNA methylation (RdDM) [54,55,56]. In *Arabidopsis*, most TEs are maintained in a transcriptionally inactive state via cytosine methylation of their DNA sequences and chromatin compaction (mainly H3K9me2), thus limiting sequence accessibility to RNA polymerase II and transcription factors. These silenced TEs can nevertheless be used to inactivate active homologous TEs in the genome in trans via RdDM [57].

Once initiated, cytosine methylation is maintained by different methyltransferases depending on their sequence contexts. In *Arabidopsis*, METHYLTRANSFERASE 1 (MET1) maintains CG methylation (mCG), and CHROMOMETHYLASE 2 and 3 (CMT2/CMT3) maintain mCHH and mCHG levels, respectively. The activity of CMT2 and CMT3 is regulated by H3K9me2, as mCHH and mCHG recruit the histone H3K9 methyl transferases to dimethylate H3K9 residues, thus forming a self-reinforcing feedback loop between CHG/CHH methylation and H3K9me2 [54], and co-localization of both epigenetic marks. DNA methylation can be erased by DNA demethylases, such as REPRESSOR OF SILENCING 1 (ROS1) in *Arabidopsis*, whose transcription is tightly controlled by methylation monitoring sequences targeted both by RdDM and ROS1 itself, and acting as a genome-wide DNA methylation sensor.

As mentioned in the previous section, environmental stress (or developmental transitions) can activate TEs and release temporarily epigenetic silencing [43,51] (see Figure 3). However, the way these epigenetic marks are alleviated during such changes, and to what extent these epigenetic silencing mechanisms characterized in *Arabidopsis*, could be generalized to other plant species, and different TEs remain to be deciphered. LTRs of the stress inducible tobacco Tnt1 retrotransposon are targeted by 24-nt siRNAs, harbor a high CG/CHG and low CHH methylation level, but there is no evidence of a differential methylation associated with stress in this respect [58].

Another example is the *Arabidopsis ONSEN* Copia retrotransposon (*ATCOPIA78*), whose transcription is induced by heat-stress in the wild-type, but is not activated in plants treated with DNA methylation inhibitor 5-azacytidine, nor in *ddm1* (DECREASE in DNA METHYLATION 1) plants with globally low levels of DNA methylation. This indicates that a decrease in DNA methylation is not sufficient to immediately release *ONSEN* silencing [59], while in contrast, transcriptional and transposition activity of another *Arabidopsis* Copia LTR-RT (*ATCOPIA93* or *Evadé*, EVD) has been observed in DNA methylation mutants [60]. Accumulation of EVD copies or of several other TEs (including class II elements) have been reported in the F8 generation of different epiRILs (epigenetic recombinant inbred lines) populations, including the methylation-defective *met1* (for EVD) or *ddm1* mutants (for several TEs) crossed to wild-type accessions [60,61]. TE reactivation in these epiRILs seems to be progressive across several generations, despite the fact that DNA methylation pathways are restored after the first F1 cross. Does the first cross represent a type of (epi)genomic stress? Why do such reactivations occur in some epiRILs and not in others? Why are several generations needed before transposition occurrence? Recent studies started addressing these questions [62,63], but the mechanistic basis remains poorly understood and restricted to just a few examples in *Arabidopsis*.

Another recent work suggests that the methyltransferase CMT3 has an unexpected role in promoting *ONSEN* transcription by competing with CMT2, and thus preventing excessive CHH methylation marks and thus H3K9me2 in *ONSEN* sequences [64]. Taken together, the response of TEs following a given stress (or developmental signal) is complex. It may depend on a particular family or individual element, the presence of regulatory motifs in the TE sequence, or to the chromatin environment that allows the recruitment of regulatory factors to the promoter of the element [52]. For example, the linker histone H1 governs targeting of the nucleosome remodeler DDM1 to heterochromatic regions in order to provide access to DNA methyltransferases that participate in the RdDM pathway [65].

### 3.3. Transposable Element Biology

Unlike viruses, endogenous TEs must replicate in the germline in order to propagate within a population, and have to preserve the reproductive fitness of their host for their long-term maintenance. They need to develop self-control strategies to maintain a basal activity, in order to limit the deleterious effects of their transposition and avoid triggering epigenetic silencing. Copy number-related TE transcriptional silencing has been reported for different plant retrotransposons introduced into heterologous system, such as *Arabidopsis* plants transformed with autonomous tobacco Tto1 and Tnt1 LTR-RTs [66,67], with a transcriptional activity recorded at a threshold of four copies for Tnt1 and 2 for Tto1. Similarly, approximately 40 copies of the retrotransposon EVD seem to be required to achieve TGS in *Arabidopsis* [62]. Interestingly, active copies of EVD are preferentially transmitted through pollen [68], perhaps because this element is not expressed in this developmental stage [69], thus escaping epigenetic silencing.

Although not demonstrated in plants (to the best of our knowledge), plant TEs could also limit their activity by the production of inactive proteins interfering with active ones during the transposition cycle as shown for the yeast Ty1 retrotransposon, *Drosophila* P element, or the horn fly Himar1 mariner TIR transposon [70,71,72].

Another strategy to limit the deleterious effects of TE transposition is insertion specificity. TE distribution is the result of the balance between element integration site preference and selection [73]. In plant genomes, the TE genomic distribution differs depending on TE superfamilies or even family/lineages. For example, Ty3/Gypsy elements belonging to the CRM lineage/family (Table 1) and some LINEs retrotransposons are found mainly in centromeres (presumably due to the presence of a specific chromodomain linked to their integrase that probably interacts with the centromeric CENH3 protein and orientates their distribution). In contrast, other Ty3/Gypsy and most of Ty1/Copia LTR-RTs are distributed in blocks or scattered along chromosomes; *Brassica* MITEs are preferentially found in gene-rich regions [74,75]. Centromeric and pericentromeric regions, as well as knobs, are “TE islands” harboring high TE density with nested TEs and constitutive heterochromatin that is epigenetically maintained. Nesting of plant LTR-RTs in such regions is not random, as it targets preferentially older copies of the same family in the 3′-UTR, and seems to be sensitive to DNA sequence, secondary structure and chromatin environment [75], as previously shown for other eukaryotes [73]. Recently, the role of the histone variant H2A.Z (incorporated preferentially in hypomethylated sequences and enriched in environmentally responsive genes in plants) in guiding integration of some Copia LTR-RTs has been evidenced in *Arabidopsis* [61,76].

### 3.4. Other Key-Players of Plant Genome Architecture

Besides the cytosine methylation and histone modification that have been already discussed, there are other biological features able to shape plant genomes and influence TE activity in plants. This includes the reproductive system (sexual/asexual), ability for hybridization and polyploidy (whole genome duplication), and the temporal dynamics of genome fractionation and the sequence elimination process through evolutionary time [77]. Genomic investigations conducted since 2000 revealed that seed plants experienced at least one round of whole genome duplication (WGD)/polyploidization [78]. Often considered as an ‘evolutionary dead end’, WGDs are now recognized as a short-term adaptive process, notably to stimulate genetic and epigenetic changes and rescue plant fitness under environmental stress, by means of DNA elimination, chromosome restructuring, or chromatin modifications [79,80].

One attractive and long-standing hypothesis is that TEs may play a central role in a situation of genomic shock triggered by hybridization/polyploidy [81]. Taking advantage of a disturbed context for epigenetic control, TEs may actively transpose and increase mutation rates or changes in gene regulation depending on the insertion sites. In such circumstances, TEs could also mediate genomic restructuring by inter-element recombination, chromosomal rearrangements and sequence elimination, as part of diploidization processes that return a polyploid genome to a diploid-like genome, such as sequence loss (duplicate genes, chromosomes or repetitive DNA), gene silencing, and altered chromosome pairing [82,83]. Analysis of TE transcription in wheat or *Arabidopsis* allopolyploid plants and their parental progenitors suggests that allopolyploidization can indeed activate some TE transcription [84,85], but this activation is not always followed by transposition and seems to be dependent on the TE family or plant species.

TE-related genomic changes have been thoroughly studied in several recent natural and synthetic allotetraploids of wheat, Brassica, Aegilops, or Nicotiana [84,86,87,88]. In these examples, no or minimal transposition events have been reported after genome merging and first generation, supporting the maintenance of active repressive mechanism of TE mobilization in the very first generation. By contrast, many TE-associated restructurings of parental insertions were observed as soon as F1 and shared across independent hybrids, as evidenced in Brassica, Aegilops or Nicotiana [87,88,89], suggesting non-random restructuring.

The study of different reciprocal plant hybrids/allopolyploids also evidenced that early genome changes often predominantly target one specific subgenome [88,89,90]. The underlying mechanisms and their timing through the polyploidization process are still not defined and may be species- or line-specific, but should involve epigenetic reprogramming, implying DNA/histone methylation, and siRNA and chromatin changes in the merged genomes [91]. If parental TE loads are divergent, the viability of the resulting hybrid will depend on the efficiency of the parental TE-derived siRNAs to control a TE population [92]. TEs can thus contribute to reproductive isolation and species formation, as recently demonstrated for the role of TE-derived small RNAs in pollen that trigger interploidy hybridization barriers in *Arabidopsis* [93,94]. Insertion profile comparison of TEs in three recent *Nicotiana* allopolyploids that arose from hybridization between increasingly divergent diploid parental species also show that quantitative imbalances between parental TE loads positively correlates with TE mobilization in the resulting allopolyploids [88]. This work highlighted TE quantitative imbalance among progenitors as an important factor for TE evolution in allopolyploids and supports the genome shock hypothesis proposed by McClintock.

## 4. Conclusions

Genomic TE content could be variable according to the species considered, and sometimes even to the genotype considered, such as for asian rice subspecies or maize lines (for a review, see [95]). TE composition and abundance results from a balance between counteracting forces, i.e., TE amplification and copy loss. TE dynamics are also impacted by host regulations and response to stress, and thus may reflect the evolution and adaptation of their host to the environment. Although potentially detrimental as a consequence of their transposition activity and repetitive nature, TEs evolved as sensors of external changes and drivers of plant evolution.

Comparison of eukaryotic genomes revealed that plants (angiosperms) have more plastic and less compartmentalized genomes than animals (mammals) [96,97]. This higher genome plasticity can be related to plant specific developmental features.

The first one is the alternation over generations of sporophytic (haploid) and gametophytic (diploid) states, coupled with double fertilization. During plant life cycles, genomes must be able to function in many different ploidy contexts—haploid (gametophyte), diploid (zygote), triploid (endosperm) contexts in diploid plants, and even higher ploidy levels in endoreduplicated cells or polyploid plants.

The second plant specificity is the absence of a sequestered germ line. As gametes differentiate late in plant development, hundreds of cell divisions take place between two zygote generations, offering many opportunities for mutations (as transpositions) to occur and to be transmitted to the offspring [96].

A third difference between plants and animals relies on the sessile nature of plants. In order to survive to stressful conditions, they might promote genomic restructuring and plasticity to generate genetic, biochemical and phenotypic novelties to adapt to external changes. By allowing transient TE silencing release during stress events, sessile plants developed a tool to generate moderate TE-mediated genomic changes which could be selected (or not) for plant adaptation to a changing environment.

Such a tightly controlled TE regulation is perhaps one of the reasons for the evolutive success of Angiosperms. Through epigenetic regulation, TEs are also thought to be involved in the maintenance of genome structure and processes, such as heterochromatin formation or reproductive isolation. Far from just being ‘junk DNA’, TEs are considered as a ‘blessing curse’ [98], and as major drivers of genome evolution and adaptation to environmental changes, even if only a part of their regulation mechanisms is understood. Each genome structure can also be considered as the result of a specific (unique) alternation of TE bursts and whole genome duplication (auto- or allopolyploidization), shaping accordingly silencing mechanisms and chromatin structure, and giving unique genomes [99,100]. Only the growing characterization of both TEs and epigenetic pathways in many plant species will help us to better understand the co-evolution of both partners.

## Figures and Tables

**Figure 1 biology-11-00488-f001:**
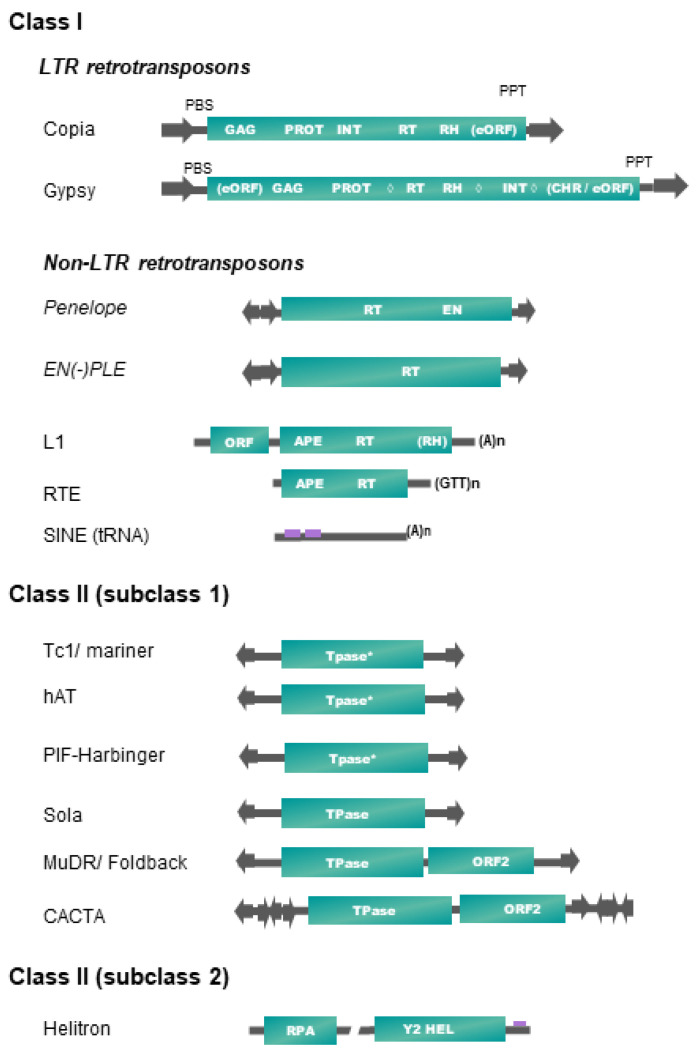
Structure and organization of plant transposable element superfamilies (adapted from [16]). Schemes are not to scale. Protein coding domains: APE = apurinic endonuclease, CHR = chromodomain, EN = endonuclease, GAG = capsid protein, HEL = helicase, INT = integrase, PROT = proteinase, RH = RNAse H, RPA = replication protein A, RT = reverse transcriptase. eORF = extra open reading frame (unknown function), Tpase = transposase (* with DDE motif), YR = tyrosine recombinase, Y2 = YR with YY motif, ◊ = different possible locations of an additional cellular-like ribonuclease H (aRH) specific of the Tat lineages (see Table 1). Optional protein-coding domains only present in some superfamily lineages are indicated in brackets. Some structural features are also represented. Terminal repeats in the same or reverse orientation are indicated by black arrows, and purple rectangles refer to diagnostic sequences present in non-coding sequences. Specific base termination of some TEs are also indicated. PBS = primer binding site, PPT = poly purine tract. Interrupted line in Helitron representation means that the region may contain one or more additional ORFs.

**Figure 2 biology-11-00488-f002:**
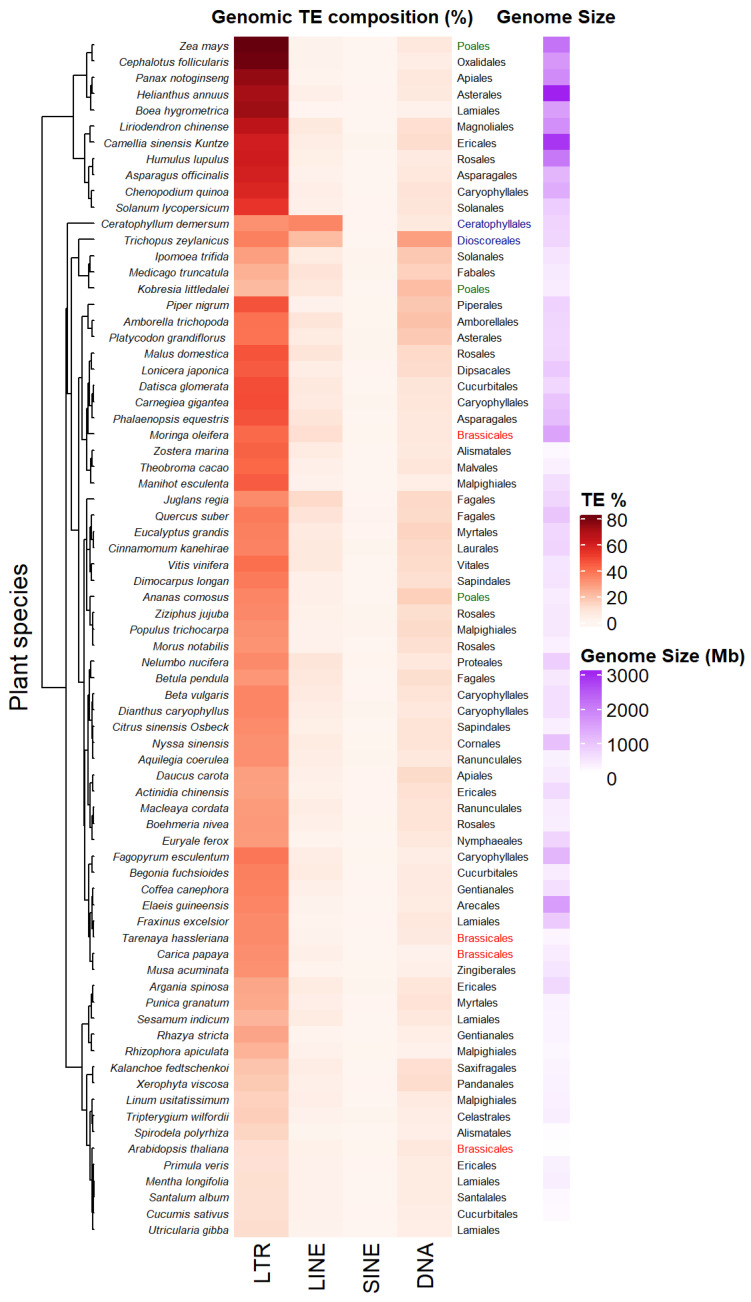
Transposable elements (TE) profiles in some land plant genomes. Species are clustered according to their TE profiles. TE percentages, plant orders and genome size estimations from 74 land plant species have been collected from [35] (data collected in Supporting information, Tables S1 and S2 from [35]). Some plant orders as Poales or Brassicales have been highlighted in colors (green and red respectively) in order to underline the diversity of TE composition between species belonging to the same plant order. Plant belonging to different orders as Dioscoreales and Ceratophyllales (in blue) can share close TE composition.

**Figure 3 biology-11-00488-f003:**
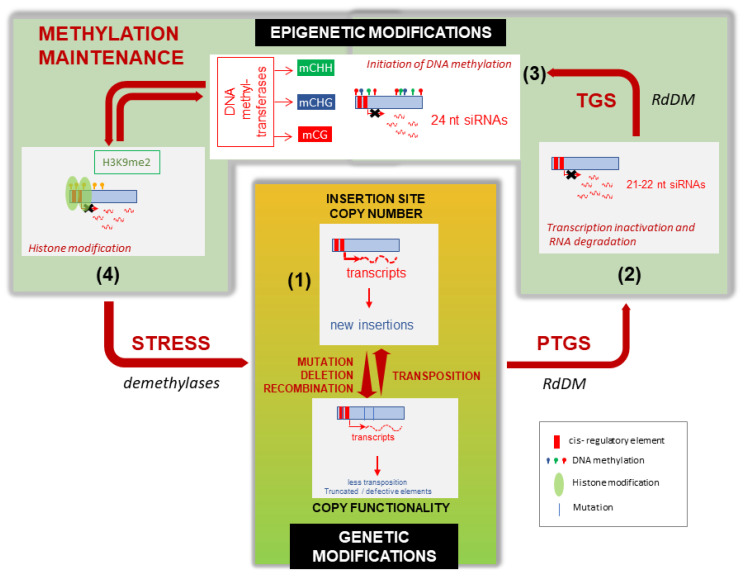
Different levels of TE transposition control in plants. (1) The activity of a new transposable element (TE) landing in a genomic place will depend on the presence of activating cis-regulating elements sensitive to stress, on the presence of similar copies, as well as insertion site preference (see text). Over time, TE copies are subjected to genetic modifications (mutations, deletions, recombinations) that will decrease copy functionality and number, as well as transposition ability. (2) High transcript abundance and presence of aberrant mRNA of inserted copies of a given TE trigger host post-transcriptional gene silencing (PTGS) that will selectively inactivate and degrade TE transcripts via the production of small interfering 21–22-nt RNAs. (3) Such inactivation is relayed by transcriptional gene silencing (TGS), relying on the production of 24-nt siRNA that will induce cytosine methylation in the three contexts, CHG, CHH and CG, via different DNA methyl-transferases (see text). (4) DNA methylation is actively maintained over cell generations and time by histone methylation (mainly histone H3 lysine 9 dimethylation) and chromatin modification by a retro- feedback mechanism involving PolV non-coding transcripts (see text). Such epigenetic modifications can be removed by cell stress, modifying the epigenetic marks and allowing transient TE activity.

**Table 1 biology-11-00488-t001:** Plant transposable element (TE) classification compiled from [16] with updates from [23] for Copia lineages, ref. [20] for Gypsy LTR retrotransposons (LTR-RTs), ref. [24,25,26] for *Penelope*-like elements (PLEs), ref. [27] for long interspaced nuclear elements (LINEs), ref. [28,29] for short interspaced nuclear elements (SINEs), and [30,31] for Sola elements.

Class	Order (Non-Autonomous TE Name)	Superfamily	Family/Lineage	Plant Family Examples
Class I	LTR-Retrotransposons	Copia	Osser	*Volvox canteri* Osser
(retrotransposons)	(LARD)		Bryco	representatives in moss species
	(TRIM/SMART)		Lyco	representatives in clubmosses species (*Lycopodiaceae*)
			Gymco-I	representatives in gymnosperms species
			Gymco-II	representatives in gymnosperms species
			Gymco-III	representatives in gymnosperms species
			Gymco-IV	representatives in gymnosperms species
			Ale/Retrofit	*Oryza longistaminata* Retrofit, *Oryza sativa* Hopscotch
			Ivana	*Oryza sativa* Oryko1-1 and Ilona, *Hordeum vulgare* HORPIA, *Nicotiana tabacum* Queenti
			Ikeros	*Zea mays* Sto-4
			Tork	*Nicotiana tabacum* Tnt1, Tto1 and Tnt2, *Solanum lycopersicum* Tork4, *Ipomea batatas* Batata
			Alesia	low copy number representatives in many Angiosperms, close to the Ale lineage
			Angela	*Triticum aestivum* Angela, *Oryza sativa* RIRE1, *Hordeum vulgare* BARE1
			Bianca	*Triticeae* Bianca, *Arabidopsis thaliana* RomaniAT5
			SIRE/Maximus	*Solanum lycopersicum* ToRTL1, *Zea mays* Opie-2, *Glycine max* SIRE1
			TAR	*Oryza* spp. Houba and Osr-1, *Arabidopsis thaliana* ATcopia95
		Gypsy (Chromovirus)	Galadriel	*Solanum esculentum* Galadriel, *Musa* Monkey, Tntom1
			Tekay	*Hordeum vulgare* Bagy-1, *Arabidopsis thaliana* Legolas Peabody, *Oryza sativa* RIRE3, *Lilium henryi* Del
			Reina	*Zea mays* Reina, *Arabidopsis thaliana* Gloin or Gimli
			CRM	*Zea mays* CRM (centromeric retrotransposon of maize), *Beta vulgaris* Beetle1, *Oryza sativa* RIRE7
		(Non-chromovirus)	Phygy	*Phycomitrella patens* Chr21 (4035670,4045566)
			Selgy	*Selaginella moellendorffii* LTR-RT
			Athila	*Arabidopsis thaliana* Athila4-1, Diaspora, *Hordeum vulgare* Bagy-2
			TatI	*Selaginella moellendorffii* LTR-RT
			TatII	*Picea abies*, *Picea glauca* LTR-RTs
			TatIII	*Picea abies*, *Picea glauca* LTR-RTs
			Ogre/TatIV + TatV	*Pisum sativum* Ogre
			Retand/TatVI	*Zea mays* Cinful-1, *Arabidopsis thaliana* Tat4-1, *Oryza sativa* RIRE2, *Sorghum bicolor* RetroSor1, *Silene latifolia* Retand
	Non-LTR retrotransposons PLE	*Penelope*/*Poseidon*		*Pinus taeda* (loblolly pine) and *Picea abies* (Norway spruce) *Dryad* PLEs by horizontal transfer
		EN(-)PLE		*Selaginella moellendorffii* spike moss, *Pinus taeda* and *Picea abies* EN(-)PLEs
	LINE	L1	Llb	sweet potato Llb, *Beta vulgaris* BvL1
			LINE-CS	*Cannabis sativa* LINE-CS, *Beta vulgaris* Belline2, Belline5
			BNR	*Beta vulgaris* Belline1/BNR
			PUR	*Carica papaya* L1-26_Cpa, *Solanum tuberosum* L1-3_Stu, *Vitis vinifera*
			Cin4	*Zea mays* Cin4
			Karma	*Oryza sativa* Karma
			nubo	*Oryza sativa* LINE-1 or OSLINE1-4, Zea mays L1-2_ZM
		RTE	plant RTE	*Malus x domestica* RTE-1_Mad, *Solanum tuberosum* RTE-1_Stu
	SINE	tRNA		*Nicotiana tabacum* TS, Au, Solanales SolS-II, Brassicale BraS-I, SB families, mainly found in Angiosperm
Class II *Subclass 1*	TIR (MITE)	Tc1-Mariner		Stowaway (MITE): *Sorghum bicolor* Stowaway, *Brassica* BraSto
		hAT		*Zea mays* Ac/Ds, *Antirrhinum majus* Tam3, *Nicotiana tabacum* Slide
		Sola		*Physcomitrella* Sola1, found also in *Capsicum annuum* and *C. baccatum*
	(MULE)	MuDR-Foldback		*Zea mays* Mu, MULEs
		PIF-Harbinger		*Zea mays* PIFa, *Oryza sativa* Pong; Tourist (MITE): mPing/Ping; mPIF/PIFa
		CACTA		*Zea mays* En/Spm, *Arabidopsis thaliana* CAC1, *Antirrhinum majus* Tam1, *Petunia hybrida* PsI
*Subclass 2*	Helitron	Helitron		*Oryza sativa*, *Arabidopsis thaliana* AthE1 Atrep, *Ipomoea tricolor* Hel-It1

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
