# Peer review of "Specificities and Dynamics of Transposable Elements in Land Plants"

_biology, 2022, doi:10.3390/biology11040488_

Round 1

Reviewer 1 Report

The manuscript is clear and provides an updated view of transposon biology in plants, suggesting hypotheses regarding genome evolution.

I believe some references should be updated; I found two or three cited reviews that were updated in the last three years (e.g. Feschotte et al., 2007).

Lines 85-92. The TE diversity and classification can be improved; the authors refer to LTR and non-LTR TEs, but there is no clear explanation regarding the differences and probable origins between LTR and non-LTR TEs. 

Table 1 and Figure 1 should also make the distinction.

Lines 171 - 215: this part lacks organization, again regarding LTR and non-LTR TEs. The authors seem to focus on SINEs and LINEs for no apparent reason and leave LTR TEs, which appear to be more diverse, in the background. The text ahead does not make this distinction. I don't know the exact reason why this part was organized like this. 

Line 223. I think instead of genus it should be genera, as the authors are talking about many different plants. 

Line 260: "depends on", instead of "depends of"

Line 396. Extra period

Line 435. "Part 3.4. Other key players of plant genome architecture" The authors left out long non-coding RNAs as part of the epigenetic control machinery, which are very much related to transposon control.

Wang, D., Qu, Z., Yang, L., Zhang, Q., Liu, Z. H., Do, T., ... & Zhu, J. K. (2017). Transposable elements (TE s) contribute to stress‐related long intergenic noncoding RNA s in plants. The Plant Journal, 90(1), 133-146.

Fort, V., Khelifi, G., & Hussein, S. M. (2021). Long non-coding RNAs and transposable elements: A functional relationship. Biochimica et Biophysica Acta (BBA)-Molecular Cell Research, 1868(1), 118837.

Hou, J., Lu, D., Mason, A. S., Li, B., Xiao, M., An, S., & Fu, D. (2019). Non-coding RNAs and transposable elements in plant genomes: emergence, regulatory mechanisms and roles in plant development and stress responses. Planta, 250(1), 23-40.

Regarding the final part, I humbly suggest these two articles for further enrichment of the discussion

Springer, N. M., Lisch, D., & Li, Q. (2016). Creating order from chaos: epigenome dynamics in plants with complex genomes. The Plant Cell, 28(2), 314-325.

Zhao, M., Zhang, B., Lisch, D., & Ma, J. (2017). Patterns and consequences of subgenome differentiation provide insights into the nature of paleopolyploidy in plants. The Plant Cell, 29(12), 2974-2994.

Author Response

We are very thankful to both Referees for their positive evaluation of our manuscript. Please find below (in blue) our point-by-point responses to each comment.

I believe some references should be updated; I found two or three cited reviews that were updated in the last three years (e.g. Feschotte et al., 2007).

We agree and included recent reviews of Wells and Feschotte (2020) (ref [19] line 90, 100) and Bourque et al (2018) (ref [18] line 92).

Lines 85-92. The TE diversity and classification can be improved; the authors refer to LTR and non-LTR TEs, but there is no clear explanation regarding the differences and probable origins between LTR and non-LTR TEs.

Table 1 and Figure 1 should also make the distinction.

We added two sentences at the end of paragraph lines 85-97 in order to give more details about the key features and probable origins of LTR and non-LTR retrotransposons. As proposed by Wells and Feschotte (2020) and Bourque et al (2018), PLEs have been included in non LTR-retrotransposons. We also modified Table 1 and Figure 1 accordingly.

Lines 171 - 215: this part lacks organization, again regarding LTR and non-LTR TEs. The authors seem to focus on SINEs and LINEs for no apparent reason and leave LTR TEs, which appear to be more diverse, in the background. The text ahead does not make this distinction. I don't know the exact reason why this part was organized like this.

We agree that such an organization does not give an accurate idea of TE diversity, but that was also not our aim in this part. In the six paragraphs between line 156 to 220 we aimed at presenting a general description of TE classification, and the length of each paragraph does not relate to the diversity and abundance of each TE type. Instead, all those aspects are discussed later in the text (lines 227-249).

Line 223. I think instead of genus it should be genera, as the authors are talking about many different plants.

Thanks. We replaced "genus" by "genera".

Line 260: "depends on", instead of "depends of"

Corrected.

Line 396. Extra period

Corrected.

Line 435. "Part 3.4. Other key players of plant genome architecture" The authors left out long non-coding RNAs as part of the epigenetic control machinery, which are very much related to transposon control.

Wang, D., Qu, Z., Yang, L., Zhang, Q., Liu, Z. H., Do, T., ... & Zhu, J. K. (2017). Transposable elements (TE s) contribute to stress‐related long intergenic noncoding RNA s in plants. The Plant Journal, 90(1), 133-146.

Fort, V., Khelifi, G., & Hussein, S. M. (2021). Long non-coding RNAs and transposable elements: A functional relationship. Biochimica et Biophysica Acta (BBA)-Molecular Cell Research, 1868(1), 118837.

Hou, J., Lu, D., Mason, A. S., Li, B., Xiao, M., An, S., & Fu, D. (2019). Non-coding RNAs and transposable elements in plant genomes: emergence, regulatory mechanisms and roles in plant development and stress responses. Planta, 250(1), 23-40.

We thank the Reviewer for this interesting suggestion, but we are not familiar with any study clearly demonstrating the role of a long non-coding RNA in TE control in plants. There are indeed examples of TE-derived lincRNA that play major roles in plant development and stress responses. However, we think this topic is not within the scope of this review.

Regarding the final part, I humbly suggest these two articles for further enrichment of the discussion

Springer, N. M., Lisch, D., & Li, Q. (2016). Creating order from chaos: epigenome dynamics in plants with complex genomes. The Plant Cell, 28(2), 314-325.

Zhao, M., Zhang, B., Lisch, D., & Ma, J. (2017). Patterns and consequences of subgenome differentiation provide insights into the nature of paleopolyploidy in plants. The Plant Cell, 29(12), 2974-2994.

We thank the Reviewer for the suggestions. These references were included in the conclusion with a new sentence in lines 521-523, where we discuss the evolution of genome structure over time, and how that could be seen as alternation of TE invasion and polyploidization events specific to each genome.

Reviewer 2 Report

The manuscript is a review of current knowledge on the classification and regulation of transposable elements present in plant genomes.

The review is complete, well organized and relevant in the field. I only have some suggestions for minor changes:

Pg.3 Line92

It says: ”based on the presence and order of specific proteins”

It may say something like: ”based on the presence and order of coding regions for specific proteins”

Pg.12 Line 226

The supplementary Tables S1 and S2 are not provided.

Pg.13 Line 279

It says: [42,41,43]

It may say: [41,42,43].

Pg.16 Line 396:

It says:  “of the element. [50].”

It may say; “of the element [50].”

Pg. 16 Lines 413-415

The paragraph refers to the production of inactive proteins interfering with active ones as a mechanism to limit TE activity. However, none of the examples are from plants. It should be made clearer that the mechanism could also exist in plants, although it has not yet been demonstrated.

Author Response

We are very thankful to both Referees for their positive evaluation of our manuscript. Please find below in blue our point-by-point responses to each comment.

It says: ”based on the presence and order of specific proteins”

It may say something like: ”based on the presence and order of coding regions for specific proteins”

This sentence line 92 has been modified according to the Reviewer's recommendations

Pg.12 Line 226 : The supplementary Tables S1 and S2 are not provided.

The two supplementary Tables S1 and S2 from which data have been extracted are contained into the Supplementary Information of the Wang et al. (2021) article (DOI = 10.1002/ece3.7222). In order to clarify the data source used for the Figure 2, we slightly changed the sentence line 231 as well as the Figure 2 description lines 224-225.

Pg.13 Line 279 .

It says: [42,41,43]. It may say: [41,42,43].

We changed the format of these three citations (lying now line 284) according to the Reviewer and Editor recommendations. [42,41,43] has been converted into [43-45] as some additional references have been added during this reviewing process.

Pg.16 Line 396:

It says:  “of the element. [50].” It may say; “of the element [50].”

The extra period after the "element" word has been removed line 401.

Pg. 16 Lines 413-415

The paragraph refers to the production of inactive proteins interfering with active ones as a mechanism to limit TE activity. However, none of the examples are from plants. It should be made clearer that the mechanism could also exist in plants, although it has not yet been demonstrated.

Thanks for the suggestion. In order to clarify this point, we modified the sentence in lines 418-421 to: "Although not demonstrated in plants (to the best of our knowledge), plant TEs could also limit their activity by the production of inactive proteins interfering with active ones during the transposition cycle, as shown for the yeast Ty1 retrotransposon, Drosophila P element, or the horn fly Himar1 mariner TIR transposon [70-72]".

Round 2

Reviewer 1 Report

The authors addressed most of the comments left and fixed the document accordingly. I still  consider lncRNAs should be included, but the manuscript works as it is right now.